# Comparison of Fetomaternal Complications in Women of High Parity with Women of Low Parity among Saudi Women

**DOI:** 10.3390/healthcare10112198

**Published:** 2022-11-02

**Authors:** Farida Habib Khan, Hend Mohammed Alkwai, Reem Falah Alshammari, Fahaad Alenazi, Khalid Farhan Alshammari, Ehab Kamal Ahmed Sogeir, Asma Batool, Ayesha Akbar Khalid

**Affiliations:** 1Department of Family and Community Medicine, College of Medicine, University of Ha’il, Ha’il 81481, Saudi Arabia; 2Department of Pediatrics, College of Medicine, University of Ha’il, Ha’il 81481, Saudi Arabia; 3Department of Pharmacology, College of Medicine, University of Ha’il, Ha’il 81481, Saudi Arabia; 4Department of Medicine, College of Medicine, University of Ha’il, Ha’il 81481, Saudi Arabia; 5Maternity and Child Hospital, Ha’il 55471, Saudi Arabia; 6William Harvey Hospital, East Kent Hospitals University NHS Foundation Trust, Ashford TN24 0LZ, UK

**Keywords:** high parity, low parity, complications, fetomaternal

## Abstract

High parity is associated with the risk of fetomaternal complications such as gestational diabetes mellitus, hypertensive disorders, maternal anemia, preterm labor, miscarriage, postpartum hemorrhage, and perinatal and preterm mortality. The objective of the study was to compare fetomaternal complications in women of high parity with women of low parity. This involved a cohort study on a sample size of 500 women who had singleton births. Data were collected from the Maternity and Child Hospital, Ha’il, Kingdom of Saudi Arabia. Participants were classified into two groups according to parity, i.e., women of low parity and women of high parity. Socio-demographic data and pregnancy complications, such as gestational diabetes, hypertension, preeclampsia, intrauterine growth restriction, etc., were retrieved from participants’ files. Participants were followed in the postnatal ward until their discharge. The results revealed that women of high parity mostly (49%) were married before 20 years of age, less educated, obese, and were of un-booked cases. Premature babies and fetal mortality are significantly high (0.000) in this group. There is a significant difference between the two groups with respect to maternal anemia, gestational diabetes mellitus, joint pain, perineal tear, miscarriage, postpartum hemorrhage, preeclampsia, vaginal tear, and cesarean section. Determinants responsible for high parity should be identified via evidence-based medicine. Public health education programs targeting couples, weight control, nutrition, and contraception would be a cost-effective strategy for reducing the risk of possible fetomaternal complications.

## 1. Introduction

The WHO defined low parity (LP) women as those having less than five pregnancies (live or stillborn) with gestation periods of ≥20 weeks, and high parity (HP) women were defined as those having five or more pregnancies (live or stillborn) with gestation periods of ≥20 weeks [1,2]. Hence, the term LP includes primipara and multipara, while HP includes grandmultipara and great grandmultipara.

HP decreased in the Western part of world and its prevalence is around 4%, but in the Eastern part of the world, HP is around 10.2%, which is more than double [2]. HP is associated with risks of maternal and fetal complications, such as gestational diabetes mellitus (GDM), hypertensive disorders, maternal anemia, preterm labor, miscarriage, postpartum hemorrhage, congenital malformations, macrosomia, and perinatal and preterm mortality [3,4,5]. Maternal age must be considered as a confounder while interpreting the risk of maternal and neonatal complications in HP women [2,4]. 

Regarding fetal complications, the latest study conducted by Cao J. et al. in China documented associations between advanced maternal age and risk for fetal chromosomal abnormalities, perinatal mortality, low birthweight babies, and preterm delivery [6]. Khalil et al. performed a cohort study in 2013 and reported a higher risk of premature babies among women with advanced maternal ages, i.e., 35 years and above (OR 1.46, 95% CI 1.27–1.69) [7]. In addition, Leader et al. in a large recent systematic review reported higher rates for low gestational age (birth weight below 10th percentile) infants among women aged above 35 years (OR 1.16, 95% CI 1.06–1.27). The authors have reported that poor oxygen exchange may be the underlying factor [8].

Regarding maternal risks in women of high parity, the incidence of molar pregnancy increases at age 35 years onward due to the hypothesis that with older maternal age, there is a greater chance of fertilizing an abnormal oocyte [9,10,11]. A retrospective study performed by Khalil and coworkers described increased gestational diabetes mellitus (GDM) incidences of 1.62 (95% CI 1.43–1.83, *p* < 0.001) and 2.1 (95% CI 1.74–2.55, *p* < 0.001) in women of advanced maternal age compared to women under the age of 35 years [7]. The risk of GDM remains higher in older ages, even after adjusting for confounding variables such as ethnicity and obesity. A similar study reported reductions in insulin sensitivity and the deterioration of pancreatic B-cell functions as main reasons for the increased incidence of GDM with age [9,11]. Advanced age is a known risk factor for hypertension due to endothelial damage, which increases with age. It is thus reasonable to expect higher rates of chronic hypertension, as well as an increased incidences of gestational hypertension and pre-eclampsia among women of age 35 years and above [10,11,12,13]. Usta et al. reported that advanced maternal age has been associated with increased intrapartum maternal morbidity and interventions [11]. 

The latest study conducted by Cao J. et al. in China on the prevalence of fetomaternal complication among women of advanced ages documented associations between advanced maternal age and the risk for fetal chromosomal abnormalities, perinatal mortality, low birth weight babies, preterm delivery, and decreased fecundity [6]. Other factors contributing to its prevalence are beliefs, norms, and illiteracy, which are the main determinants that limit contraceptive use [5,10]. HP is a burden to health care as well as to families’ economical systems [6,9]. It has been reported from relevant studies that most women of HP had Lower Segment Caesarean Section (LSCS) because of malpresentatmion, placenta previa, and abruptio placentae [9]. Although a scarred uterus in high-parity women increased the risk of delivery by emergency caesarean sections by 2.4-fold compared to women of low parity [12]. 

Ben-Aroya et al. performed a cohort study in 2001 in Israel on 424 grand multiparous women where intravenous oxytocin was used for the augmentation of labor. A significant increase in the rate of vacuum deliveries was observed in patients given oxytocin compared to the controls (3.5% vs. 1.4%, respectively; *p* = 0.001) [13]. However recent studies have shown that with good perinatal care, routine follow-ups, and use of family planning methods, the risk of complications in HP decreases [14,15,16,17]. Nonetheless, recent studies have shown that with good perinatal care, routine follow-ups, and use of family planning methods, the risk of complications in HP could decrease [14,15,16,17,18]. For cultural reasons, a large family size is desirable in the Kingdom of Saudi Arabia (KSA); in addition, marriage at a young age is a common practice [14,18]. Other factors contributing to the prevalence of high parity are illiteracy and norms that are stumbling blocks to contraceptive use [14,15,18]. Consequently, a high incidence of high parity is expected. 

The aim of our study was to compare fetomaternal complications in women of high parity with women of low parity among Saudi Women in the Kingdom of Saudi Arabia.

## 2. Materials and Methods

This was a cohort study conducted within a 6-month period between 1 October 2021 and 30 June 2022 on a sample size of 500. After obtaining ethical approval from the Research Deanship, University of Ha’il, data were collected from the Labor Ward of Maternity and Child Hospital, Ha’il, Kingdom of Saudi Arabia. Participants were classified into two groups according to parity. Primipara (having one birth) and multipara (having 2–4 births) constitute a group of women with low parity (LP), while grand multipara (having 5 or more births) and great grandmultipara (having 10 or more births) constitute a group of women with high parity (HP). 

Inclusion criteria included all pregnant women who delivered a single neonate at a gestation age of ≥28 weeks. 

Exclusion criteria included pregnant women with multiple gestations, illnesses that could cause adverse outcomes during pregnancy such as renal and cardiac diseases, known diabetics and hypertensive before the first pregnancy, and smokers. Women who were seriously ill to the extent of not being able to communicate were also excluded. 

Informed verbal consent was obtained from all participants prior to their participation in the study. Socio-demographic data and pregnancy complications such as gestational diabetes or hypertension (de novo hypertension alone after 20 weeks of gestation in a previously normotensive woman), preeclampsia, intrauterine growth restriction, etc., were retrieved from participants’ files. Records from the Labor Ward were used to note information about natal events (e.g., spontaneous preterm delivery and caesarean section), and birth outcomes (e.g., anthropometric birth outcomes, APGAR score in the 5th minute after delivery, congenital malformations, maturity, and newborn admission to the ICU) were noted after delivery. A newborn birth weight of <2500 g was considered as low birth weight; in addition, a low APGAR score corresponded to a score <7 in the 5th minute after delivery [6]. Participants were followed in the postnatal ward until their discharge.

Data were analyzed using SPSS software v.23.0 for Windows^®^ (SPSS Inc., Chicago, IL, USA). Differences between groups were assessed using Chi-square (χ^2^) test. All statistical tests were two-tailed, and a *p* value ≤ 0.05 was considered statistically significant.

## 3. Results

Table 1 shows the demographic profile of our respondents. Most of our respondents (49%) were married before 20 years of age and are housewives (87%). Ninety percent of respondents belonged to households with a total monthly income of less than 15,000 Saudi Riyals. Almost half of the mothers (51%) had school education up to the fifth class. 

In our study, 20% of mothers were overweight, while a large majority (72%) was obese, as shown in Figure 1.

Figure 2 shows that 20% of mothers used the barrier method followed by the withdrawal method (18%). 

Oral contraceptive pills (OCP) were used by only 5%, while other methods were negligibly used. The majority of respondents (49%) did not practice any sort of contraception (Bar with 0). Figure 3 highlights different reasons for not practicing contraception. The main reason (37%) found is the desire to have a male child.

Table 2 shows fetal complications between the two groups. It was found that women of HP had more premature babies and fetal mortality, and the difference is highly significant (*p* = 0.000). There was no difference in Apgar scores at 5 min (≥7), the rate of admission in ICU, and birth weight between the two groups (*p* = 0.787, 0.909, 0.316, respectively).

When maternal complications were compared between the two groups (Table 3), it was found that women of high parity were mostly those who were married before 20 years of age, less educated (under high school), obese, and were un-booked cases. Regarding medical complications, there was a significant difference between the two groups with respect to anemia, gestational diabetes mellitus, and joint pain (*p* = 0.001, 0.004, 0.025, respectively). Obstetrical complications were observed more among women of high parity compared to women of low parity (perineal tear, first and second trimester miscarriage, PPH, preterm delivery, preeclampsia, placenta previa, and vaginal tear), and statistically, the difference is significant. Almost all women of high parity were not taking any sort of contraception (*p* = 0.001). 

Table 4 shows that most women of high parity had LSCS (lower segment caesarean section) because of malpresentation, the failure of induction, and placenta previa. There is highly significant difference between the two groups (0.000).

## 4. Discussion

Throughout the Middle East, Africa, India, and Pakistan, a large family size is highly valued, although its consequence is high fertility [17]. 

The culture of early marriages and false beliefs that do not support the use of contraception are the main determinants responsible for an increase in the incidence of high parity in the Saudi population [14,18]. 

The current study shows that women of high parity were mostly those who were married before 20 years of age and are less educated (under high school), obese, and un-booked cases. These findings are consistent with the findings of research studies performed in Nigeria, India, and Pakistan [4,10,19,20,21,22].

Regarding maternal complication, the vast majority of women of high parity in this study were found to be anemic (Hb. < 11 gm/dl) and have GDM. Graham W and Munium et al. reported similar findings in their studies performed in Africa and Pakistan, respectively [17,21].

Previous studies have shown that essential hypertension was 8 times more frequent among women over 30 years than it was in younger women [12,14]. This finding supports the evidence that hypertension is a common complication in women of high parity because of their age determinant. TAhe study conducted by Mgaga et al. in 2013 revealed that preeclampsia and gestational hypertension were significantly associated with the age of the parturient, the authors also mentioned that preclampsia during the first pregnancy was a predictor of hypertensive complications during subsequent pregnancies [16]. Our study results also supported that women of high parity had a significant association (0.0047) with past obstetrical histories of preeclampsia.

The latest study performed by Vidiri et al. in 2022 on water birth has observed that hydrotherapy has marked physiological effects on the cardiovascular system: Shoulder-deep warm water immersion reduces blood pressure due to the vasodilation of peripheral vessels and the redistribution of blood flow [22]. Water immersion is generally considered a safe and low-cost method of pain management for women in the first stage of labor. However, during the second and third stages, there is a risk of fetal complications. 

Similarly, Fowler-Brown et al. found that the risk of diabetes in women of HP was reduced after adjustments for the maternal age and body mass index (BMI). The authors highlighted the effect of old age and increased BMI on the development of gestational diabetes mellitus [23,24].

Previous studies revealed that recurrent pregnancies and breastfeeding predispose the individual to poor maternal nutrition [25,26,27]. A poorer nutritional status in high-parity mothers occurs as a result of less parental investments. Overall, there is a trend toward the lower utilization of maternal health services as birth order increases. The majority of maternal health indicators maintained their significant negative linear relationship with parity even when controlling for poverty [28]. Chowdhury et al. documented, in a systematic review and meta-analysis in 2015, that due to recurrent breastfeeding; high-parity mothers are likely to suffer from hypocalcemia, resulting in osteoporosis and joint pain [29]. In relevant research studies, it was observed that femoral bone marrow densities significantly decreased as parity increased [25,26,29]. It is hypothesized that after the discontinuation of breastfeeding, bone resorption returns to normal while bone formation continues; therefore, there is possibility restoring bone densities after lactation [26,27].

There is a significant difference in the usage of contraception between the two groups. Our result in this issue has been supported by many national and international research findings [18,30]. The main reasons for this attitude are as follows: cultural norms against contraception, the fear of its side effects, and the desire for having a large family size and a male child [18,30].

Our study has revealed that women of high parity are more likely to have PPH, perineal tear, and miscarriage. Miyoshi et al. documented that PPH is the leading cause of maternal mortality, accounting for 27.1% of all maternal deaths [31]. Women of high parity are at risk of having PPH due to abruption placentae, placenta previa, and uterine atonia [32,33,34]. An advanced maternal age in grand and great grand multipara is also one of the risk factors [35]. Associated factors include a high rate of unmet contraceptive needs and low socio-economic statuses [17,30]. Miyoshi et al. reported that the optimal parity cut-off value for predicting PPH in vaginal deliveries was para 7, while the optimal parity cut-off value for predicting PPH in cesarean section deliveries was para 3 [31]. On the contrary, these complications are negligible in other developed countries where perinatal services are vigilantly utilized by the majority of mothers, and contraception is practiced by more than 80% of couples [19,30].

Placenta previa, the failure of induction, and malpresentation were significant reasons for caesarean sections among women of high parity, and the same findings were observed in previous studies performed by Saadia and Munium [20,21]. This observation was in contrast to research studies performed by Thekrallah et al. in which an abruptio placenta was more frequent [30], although Tikkanen et al. reported that placental abruption has associations with pre-pregnancy hypertension and eclampsia [32].

In this study, the majority of HP cases had caesarean section, which correlates with other studies [32,33,34]. The high incidence of caesarean sections in our study population could be due the choices that the patient is given to have deliveries. In contrast, the study conducted by Munium et al. found no significant difference in the proportion of caesarean section or normal delivery between the two groups (HP vs. LP women) [21].

Regarding fetal complications, women of HP were more likely to have premature babies resulting in fetal deaths compared to women of LP [10,14,18]. Premature deliveries could occur due to a weakened cervix that opens early, advanced maternal age, poor nutritional status of mother during pregnancy, and pre-eclampsia [30,35].

However, there is no significant difference in Apgar scores at 5 min between the two groups, and our findings are consistent with similar studies conducted in Pakistan [21,34]. On the contrary, some studies performed in the same region revealed low Apgar score in babies born to grandmultipara and great grand multiparas [19,20].

There are only a few studies that support the observation that high parity does not necessarily entail significant maternal, fetal, and neonatal complications in societies with high socioeconomic statuses, access to quality health care, and those that practice contraception [34,36]. Similarly, Saadia, et al. and Munium et al. found insignificant associations of fetomaternal deaths in women of high parity in societies where couples strictly follow antenatal visits and practice family planning [20,21].

## 5. Limitation of Study

As our study population was Saudi females exclusively, hence results could not be generalized especially with respect to contraception. 

## 6. Conclusions

Maternal complications were identified were anemia, miscarriage, GDM, and PPH. There was a significant association of premature deliveries with high parity. Factors responsible for adverse fetomaternal outcomes were advanced maternal age and unmet contraceptive needs. Misconceptions and social taboos about family planning and the use of contraception can only be dispelled by sensitive and sympathetic counseling with the involvement of the male partner. Determinants that result in adverse fetomaternal outcomes should be identified via evidence-based medicine. Furthermore, public health education programs targeting couples, female literacy, weight control, nutrition, and contraception would be cost-effective strategies for reducing the risk of possible fetomaternal complications.

## Figures and Tables

**Figure 1 healthcare-10-02198-f001:**
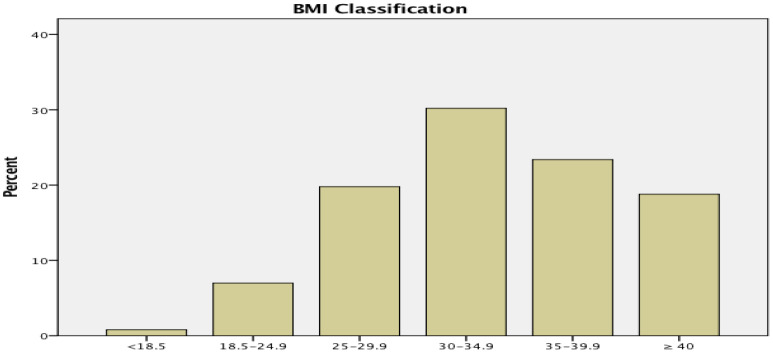
BMI (body mass index) of respondents: Underweight (<18.5 kg/m^2^)—1%; Normal (18.5–24.9 kg/m^2^)—7%; Overweight (25–29.9 kg/m^2^)—20%; Obese Class I (30–34.9 kg/m^2^)—30%; Obese Class II (35–39.9 kg/m^2^)—23%; Obese Class III (≥40 kg/m^2^)—19%.

**Figure 2 healthcare-10-02198-f002:**
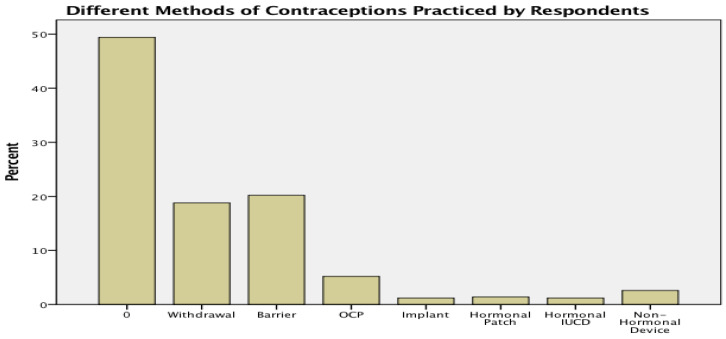
Different methods of contraception used by respondents (Abbreviation OCP stands for oral contraceptive pills; IUCD stands for intra-uterine contraceptive device; 0 stands for non-contraceptive users).

**Figure 3 healthcare-10-02198-f003:**
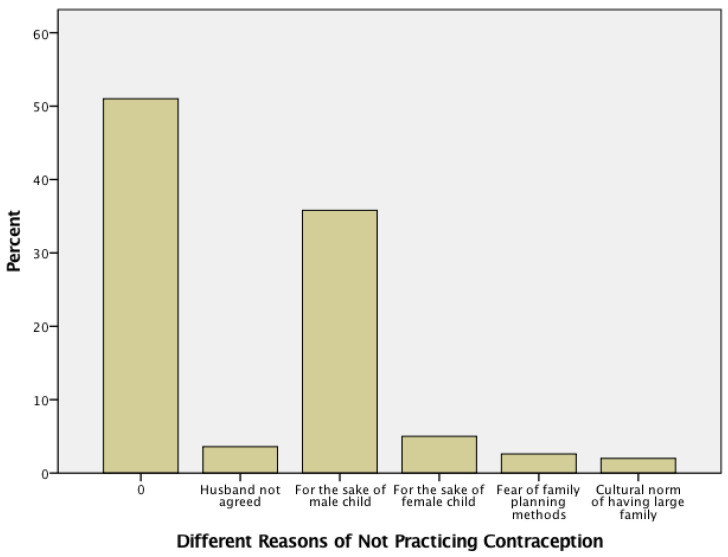
Reasons listed by respondents of not using any sort of contraception (Abbreviation 0 stands for contraceptive users).

**Table 1 healthcare-10-02198-t001:** Demographic Profile of Mothers (*n* = 500).

Variable		Frequency	%	Cumulative %
Age in years at Marriage	15–20	244	49	49
	21–30	138	28	77
	31–40	91	18	95
	41–45	8	1	96
	Missing	19	4	100
Occupation of Mother	Housewife	434	87	87
	Working	66	13	100
Monthly Household Income (Saudi Riyals)	<15,000	452	90	90
	15,000–20,000	33	7	97
	20,001–30,000	7	1.5	98.5
	30,001–40,000	6	1	99.5
	>40,000	2	0.5	100
Level of Education	Uneducated	47	9	9
	Primary	38	8	17
	Middle	172	34	51
	High School	215	43	94
	Bachelor	20	5	99
	Masters	8	1	100

**Table 2 healthcare-10-02198-t002:** Association between women of low parity and high parity with respect to fetal complications (application of Chi-Square test keeping the level of significance ≤ 0.05).

Comparing Variable	*p*-Value
Fetal Outcome (Alive/Dead)	0.000
Admission in ICU (Yes/No)	0.909
Fetal Maturity (Pre-mature)	0.000
APGAR Score at 5 min (≥7)	0.787
Birth Weight (LBW/Not LBW)	0.316

(Abbreviation ICU stands for intensive care unit; LBW stands for low birth weight).

**Table 3 healthcare-10-02198-t003:** Association between women of low parity and high parity with respect to demographic profile, medical history, past obstetrical history, drug history, and contraception (application of Chi-Square test keeping the level of significance ≤ 0.05).

Comparing Variable	Comparing Variable	*p*-Value
DEMOGRAPHIC PROFILE	Age at marriage (< and >20 years)	0.006
	Total monthly income	0.960
	(< and >10,000 Saudi Riyals)	
	Educational Level(< and >middle school)	0.050
	BMI (normal versus obese)	0.020
MEDICAL COMPLICATIONS	Anemia	0.001
	Hypertension	0.080
	Gestational Diabetes mellitus	0.004
	Cardiac Disease	0.937
	Joint Pain	0.025
PAST OBSTETRICAL HISTORY	Booked/Unbooked Case	0.028
	Caesarean Section	0.003
	Perineal Tear	0.040
	1st trimester miscarriage	0.000
	2nd trimester miscarriage	0.004
	PPH	0.010
	Placenta previa	0.050
	Pre-eclampsia	0.047
	Vaginal Tear	0.018
DRUG HISTORY	Intake of Table Folic Acid	0.005
	Intake of Table Calcium	0.253
	Intake of Table Iron	0.285
CONTRACEPTION	Usage of contraception	0.001

(Abbreviation BMI stands for body mass index; PPH stands for postpartum hemorrhage).

**Table 4 healthcare-10-02198-t004:** Association between women of low parity and high parity with respect to different reasons for LSCS (*n* = 307) (Application of Chi-Square test keeping level of significance ≤ 0.05).

Parity	Malpresentation	Failure of Induction	Placenta Previa	*p*-Value
Low Parity	15	90	13	0.000
High Parity	36	122	31	
Total	51	212	44	

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
