# Peer review of "Comparison of Fetomaternal Complications in Women of High Parity with Women of Low Parity among Saudi Women"

_healthcare, 2022, doi:10.3390/healthcare10112198_

Round 1
Reviewer 1 Report
Dear authors,
I read with great interest the manuscript, which falls within the aim of this Journal. In my honest opinion, the topic is interesting enough to attract the readers’ attention. Nevertheless, authors should clarify some points and improve the discussion, as suggested below. Authors should consider the following recommendations:
In my opinion you have to improve the paper focusing how in high parity is possible to have an increased risk oh hypertension in pregnancy and how the waterbirth can help in case of pain during the labour.
I suggest you to read ancd cite these papers
Waterbirth: current knowledge and medico-legal issues
The role of serum potassium and sodium levels in the development of postpartum hemorrhage. A retrospective study
Author Response
Dear Reviewer, thanks for your comments. I have done the needful.
Comment- How in high parity is possible to have an increased risk oh hypertension in pregnancy?
As previous studies have shown that essential hypertension was 8 times more
frequent among women over 30 years than it was in younger women [12,14]. This finding supports the evidence that hypertension is a common complication in women of high parity because of their age determinant. Study done by Mgaga et al in 2013 has revealed that preeclampsia and gestational hypertension were significantly associated with age of the parturient, authors also mentioned that preclampsia during the first pregnancy was a predictor of hypertensive complications during subsequent pregnancies[16]. Our study results also supported that women of high parity has a significant association (0.0047) with past obstetrical history of preeclampsia.
Comment- How the waterbirth can help in case of pain during the labour?
Latest Study done by Vidiri et al in 2022 on Water-birth has found that hydrotherapy has marked physiological effect on the cardiovascular system: shoulder-deep warm water immersion reduces blood pressure due to vasodilation of the peripheral vessels and redistribution of blood flow [22]. Water immersion is generally considered a safe and low-cost method of pain management for women in first stage of labor. Though in the second and third stage there is a risk of fetal complications.
Reviewer 2 Report
Manuscript ID: healthcare-1939408
Title: Comparison of fetomaternal complications in women of high parity with women of low parity
Authors: Farida Habib Khan, Hend Mohammed Alkwai, Reem Falah Alshammari, Fahaad Alenazi, Khalid Farhan Alshammari, Ehab Kamal Ahmed Sogeir, Asma Batool and Ayesha Akbar Khalid
In this short retrospective study on Saudi women, the authors presented interesting statistical analyses on maternal parity, social norms related to the use of contraception as well as the risk of cesarean section and obstetrical complications linked to increased risk of maternofetal health adversities among this specific middle-eastern women population. However, as detailed below, certain articles related to the preparation of the manuscript would have to be addressed and extensive edits are required conducive to having a scientifically sound and meaningful article of this calibre.
Major:
- Title: since this retrospective study was exclusively conducted on Saudi women, the title of this manuscript should reflect this fact (e.g., Comparison of fetomaternal complications in women of high parity with women of low parity in the Kingdom of Saudi Arabia) (or “among Saudi women”)!
- Introduction:
1- Lines 53- 54: Please elaborate on the “confounding” nature of the age factor in assessing the risk of maternal and neonatal complications in HP women
2- Line 55: please replace “it” in “its prevalence” with the actual pronoun referenced in this statement.
3- Lines 58-60: Please provide the incidence rate of Lower Segment Caesarean Section (LSCS) in women of HP referenced in this statement.
4- Lines 61- 62: Please provide the incidence rate of vacuum applications after the use of oxytocin in women of HP referenced in this phrase.
5- Lines 70-71: Please indicate the country/region of your cohort study reference in this statement.
Materials and methods:
1- Line 73: please check the grammar
2- Please provide a detailed table summarizing the inclusion and exclusion criteria used in your study.
- Discussion:
1- Lines 151- 154: truncated sentences: please check the grammar and consider rephrasing accordingly
2- Lines 155-157: Graham W et al. and Munium et al did not report your present findings- please consider rephrasing your arguments accordingly.
3- Lines 162- 163: please provide further insights and elaborate on the plausible associations between recurrent pregnancies as well as breastfeeding and predisposition to poor maternal nutrition as reported in the present manuscript
4- Lines 164- 173: should be combined in one paragraph with further insights into potential links between parity and poor maternal nutrition and joint pain in obese HP women as reported in this manuscript.
5- Lines 174- 178: should be combined in one paragraph with further insights and elaboration on the use of contraception in your cohorts.
6- Lines 179- 182: please further elaborate on the risk of PPH in your cohort of women with HP- your current argument is insufficient as presented.
7- Lines 183- 192: should be combined in one paragraph and further elaborate on potential risks and benefits of cesarean section in your cohort of HP women accordingly.
8- Lines 193- 198: please provide further insights into factors contributing to your present observations of high fetal mortality among HP women having a comparable Apgar score at 5 minutes with LP women in your study cohort.
9- Lines 205- 213: your conclusions should also reference the many maternofetal issues noted in 3-8 in the above- your current conclusions are insufficiently summarized.
Author Response
Dear Reviewer, thanks for your valuable comments. I have done the needful. Please see the attachment. The comments are in red font color while reply/edited text is in blue font color.

Reviewer 3 Report
Comparison of fetomaternal complications in women of high parity with women of low parity
Main observations:
The manuscript topic is consistent with the journal content.
The authors rightly concluded determinants that result in adverse fetomaternal outcomes should be identified through evidence-based medicine”.
LACK of LIMITATION of study (at the end of the Discussion section).
Lack of precision in units of BMI values.
Literature is relatively out of date - more than 65% are articles more than10 years old - should use more current items.
The discussion is consistent with the evidence and arguments and addresses the primary objective.
It is recommended that the entire manuscript and literature be reviewed for spelling errors.
In my opinion, this study would be a candidate for publication in your journal as an original article, with minor revisions.
Minor observations:
INSTEAD OF: W.H.O. has defined low parity (LP) women as those having less than 5 pregnancies (live or stillborn) with gestation periods of ≥ 20 weeks, and high parity (HP) women as those having five or more pregnancies (live or stillborn) with gestation periods of ≥ 20 weeks [1,2].
SHOULD BE: WHO has defined low parity (LP) women as those having less than five pregnancies (live or stillborn) with gestation periods of ≥ 20 weeks, and high parity (HP) women as those having five or more pregnancies (live or stillborn) with gestation periods of ≥ 20 weeks [1,2].
In Table 1 INSTEAD OF:
Level of Education Masters 08 1 99
SHOULD BE:
Level of Education Masters 8 1 99
In Figure 1. INSTEAD OF: BMI of Respondents. Underweight (<18.5)---------1%; Normal (18.5-24.9)-----------7%; Overweight (25-29.9)-------20%; Obese Class I (30-34.9)-----30%; Obese Class II (35-39.9)----23%; Obese Class III (≥ 40)--------19%
SHOULD BE: BMI (Body Mass Index) of Respondents. Underweight (<18.5 kg/m2)---------1%; Normal (18.5-24.9 kg/m2)-----------7%; Overweight (25-29.9 kg/m2)-------20%; Obese Class I (30-34.9 kg/m2)-----30%; Obese Class II (35-39.9 kg/m2)----23%; Obese Class III (≥ 40 kg/m2)--------19%
NO EXPLANATION OF ABBREVIATIONS UNDER FIGURES #1-3 AND TABLES #1-3.
Author Response
Dear reviewer, thanks for your valuable comments. I have done the needful. Please see the attachment. The comments are in red font color while reply/edited text is in blue font color.

Round 2
Reviewer 2 Report
Thank you for your revisions.
Please remove the word "joint" in line 303 as it lacks scientific merits.
Author Response
Dear Reviewer, thanks for your valuable comments and formatting. It was very helpful to me. As per your suggestion, I have removed the word " JOINT" from line number 303. I hope this time the manuscript will be according to the requirements of healthcare.